# Pick-and-Draw: Training-free Semantic Guidance for Text-to-Image Personalization

Henglei Lv[*]
Institute of Computing Technology,
Chinese Academy of Sciences
Beijing, China
lvhenglei22s@ict.ac.cn

Jiayu Xiao[*]
Institute of Computing Technology,
Chinese Academy of Sciences
Beijing, China
jiayu.xiao@vipl.ict.ac.cn

Liang Li[†]
Institute of Computing Technology,
Chinese Academy of Sciences
Beijing, China
liang.li@ict.ac.cn

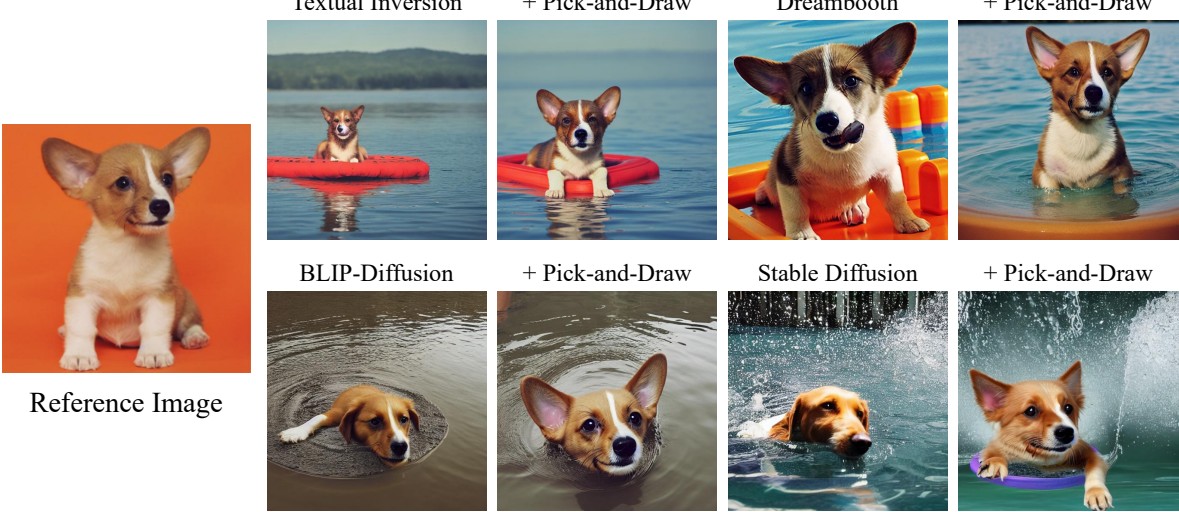

**Figure 1: Given a single reference image, Pick-and-Draw consistently improves identity consistency and image-text alignment over various personalization methods, including Textual Inversion, DreamBooth, and BLIP-Diffusion. The text prompt is "A dog in water". Additionally, directly applying Pick-and-Draw on vanilla Stable Diffusion also produces acceptable outcomes.**

## Abstract

Diffusion-based text-to-image personalization has achieved great success in generating user-specified subjects in various contexts. However, finetuning-based methods often suffer from model overfitting, leading to reduced generative diversity, particularly when the provided subject images are limited. To address this issue, we introduce Pick-and-Draw, a training-free semantic guidance approach that enhances identity consistency and generative diversity. Our method comprises two key components: appearance-picking guidance and layout-drawing guidance. In the appearance-picking phase, we create an appearance palette from visual features of the reference image, selecting local patterns to maintain consistent subject identity. In the layout-drawing phase, we use a generative template from the base diffusion model to sketch the subject shape and scene outline, leveraging its strong image prior to produce diverse contexts based on various text prompts. Pick-and-Draw can be seamlessly integrated with any personalized diffusion model and requires only a single reference image. Both qualitative and quantitative evaluations demonstrate that our approach significantly improves identity consistency and generative diversity, establishing a new Pareto frontier in the balance between subject fidelity and image-text alignment.

[*] Both authors contributed equally to this work.

[†] Corresponding author.

## CCS Concepts

• **Computing methodologies** → *Computer vision problems.*

## Keywords

Text-to-Image Personalization; Diffusion Model

**ACM Reference Format:**

Henglei Lv, Jiayu Xiao, and Liang Li. 2024. Pick-and-Draw: Training-free Semantic Guidance for Text-to-Image Personalization. In *Proceedings of the 32nd ACM International Conference on Multimedia (MM '24), October 28-November 1, 2024, Melbourne, VIC, Australia.* ACM, New York, NY, USA, 9 pages. https://doi.org/10.1145/3664647.3680658

# 1 Introduction

Recent large-scale diffusion models [7, 21, 22, 24] demonstrate remarkable capability in text-to-image generation. Trained on billions of image-text pairs collected from the Internet, these models are competent to synthesize high-quality and diverse images conditioned on textual inputs. Owing to unprecedentedly strong image priors, text-to-image diffusion models are successfully applied to various downstream tasks, including image editing [2, 8, 12], inpainting [34, 37], augmentation [9, 26], style transfer [29, 35, 39] and controllable generation [31, 32, 38].

As a newly emerged task, text-to-image personalization aims to reason over specified subjects in assorted contexts. It requires the model to mimic the appearance of a subject given a reference image set and synthesize the same subject in different contexts. Many works [10, 16, 23] are proposed and have achieved impressive results. However, these methods still suffer from severe mode collapse in data-scarce scenarios where only a few reference images are available, and the diffusion network tends to simply memorize the few reference samples during the fine-tuning process. As a result, the model struggles to follow text instructions and synthesize subjects of different views, poses, and backgrounds. To mitigate this problem, some works [15, 23] leverage a regularization set to preserve the image priors of the original diffusion model, while others propose to fine-tune a subset of model parameters [15] or introduce extra low-rank adaptors [13]. These approaches help preserve the innate capabilities of the model, yet require extensive empirical hyperparameter tuning to obtain delicate results, and optimal hyperparameter configurations may vary across different subjects. Balancing the identity consistency and context diversity of generative outcomes remains a challenging problem.

To this end, we propose Pick-and-Draw, training-free semantic guidance on text-to-image personalization methodologies, aiming to boost identity consistency while maintaining the ability of diverse context synthesis. In general, our approach consists of two components: appearance-picking guidance and layout-drawing guidance. (1) As for appearance-picking guidance, we feed the inverted latent of the reference image into the deep generative network and extract a visual feature set as a palette, from which we pick "color" for generating the specified subject. Specifically, we first adopt the layer-wise cross-attention maps corresponding to the specific subject, and threshold them to obtain salient binary masks. We then leverage the masks to extract the feature vectors within the object regions. Subsequently, we minimize the Unidirectional Relaxed Earth Mover Distance (UREMD) between the above feature vectors of the reference image and the generated image at each denoising step, to aid the model in better capturing the appearance cues of the new concept during the generation process. (2) As for layout drawing guidance, we borrow the subject shape and scene contour generated by the original diffusion model as a template. Then we imitate the outline to enable diverse posture and context synthesis of the new concept for a personalized model. This process aligns the subject's posture and scene context with the template to ensure diverse generated outcomes. The overall pipeline of our approach bears resemblance to the painting process of picking colors from a palette, drawing outlines based on a template, and

subsequently applying colors to finalize the entire painting. In this sense, we term our method Pick-and-Draw.

Pick-and-Draw is a training-free plug-and-play semantic guidance approach developed for boosting text-to-image personalization, applicable to various personalized models including Texual Inversion [10], DreamBooth [23], and BLIP-Diffusion [16]. Our method consistently improves personalized methods' identity consistency and generative diversity, pushing the trade-off between image fidelity and textual alignment to a new Pareto frontier. Moreover, we surprisingly find that directly applying Pick-and-Draw to vanilla Stable Diffusion [22] also yields favorable outcomes.

To summarize, we make the following key contributions:

- We propose Pick-and-Draw, a training-free semantic guidance approach to enhance identity consistency and generative diversity for text-to-image personalization models.
- We demonstrate quantitatively and qualitatively that Pick-and-Draw consistently improves identity preservation and diverse context synthesis of various personalized models, pushing the trade-off between subject fidelity and image-text fidelity to a new Pareto frontier.
- We find that directly applying Pick-and-Draw to vanilla Stable Diffusion yields surprisingly favorable outcomes, which may potentially inspire research on training-free single-image personalization.

# 2 Related Work

## 2.1 Text-to-image diffusion

Diffusion models are a class of generative models that learn image distributions through sequential denoising. A diffusion model consists of a diffusion process and a reverse process. Given an initial image $x_0$, the diffusion process gradually adds Gaussian noise $\epsilon_t$ in $T$ time steps until $x_0$ is diffused into $x_T$ which conforms to a Gaussian distribution. The reverse process aims to recover $x_0$ given $x_T$ by training a denoiser $\epsilon_\theta$ that predicts the noise $\epsilon_t$ given timestep $t$ and the noisy image $x_t$ using diffusion loss:

$$L(\theta) = \mathbb{E}_{x_0,t,\epsilon_t \sim \mathcal{N}(0,1)}([||\epsilon_t - \epsilon_\theta(x_t,t)||^2]). \tag{1}$$

Stable Diffusion (SD) [22] is a powerful text-conditioned latent diffusion model that performs diffusion in the latent space $Z$ instead of the pixel space $X$ and injects text condition into the diffusion process, allowing flexible conditional generation.

## 2.2 Energy functions in diffusion models

From a score-based perspective, each step in the reverse process in a diffusion model can be seen as an estimate of a score function $\nabla_{x_t} \log p(z_t)$ [25]. Given an external condition $y$, diffusion models generate conditional samples from $p(z_t|y) \propto p(z_t)p(y|z_t)$. The first term $p(z_t)$ corresponds to the unconditional score function, and the second term $p(y|z_t)$ is equivalent to an energy function $\mathcal{E}(z_t; t, y)$. Numerous energy functions have been proposed and used in various tasks, including classifier guidance [7], CLIP scores [19], and attention penalties [6, 8, 31, 33, 36]. In this sense, we propose two energy functions to boost identity consistency and maintain context diversity, respectively.

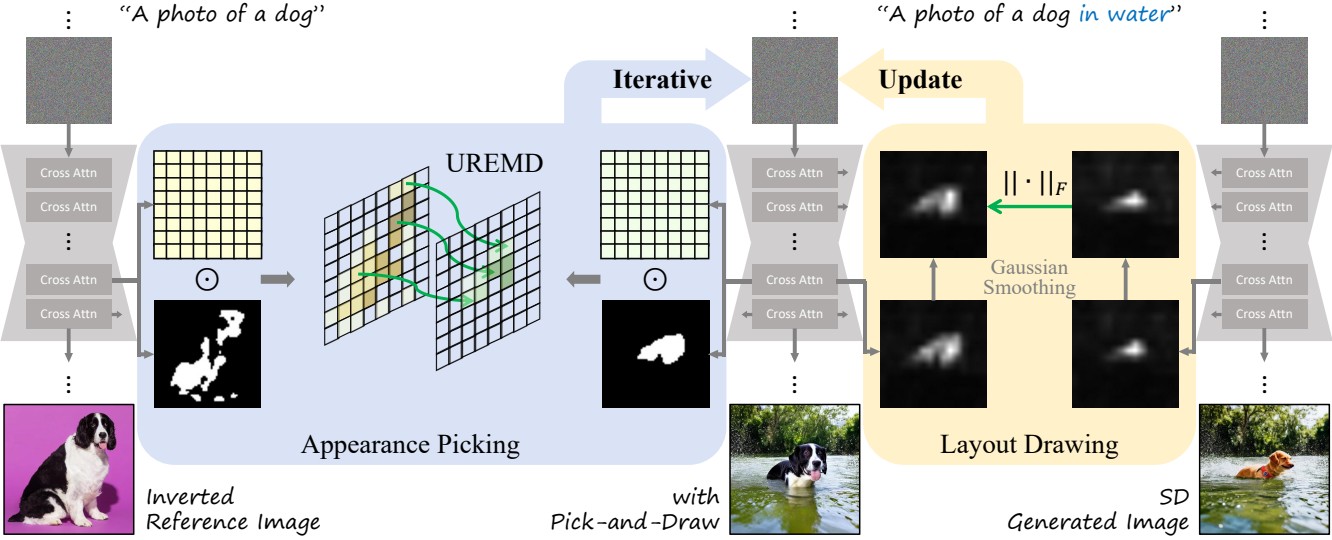

**Figure 2: Overall pipeline of our proposed Pick-and-Draw. We iteratively refine the generative outcomes via appearance picking and layout drawing, which is achieved by optimizing a designed score function. In appearance picking, we pick saliency-aware features from certain cross-attention decoder layers and transfer the appearance cues by minimizing the Unidirectional Relaxed Earth Movers Distance (UREMD), aiming to boost identity consistency. For layout drawing, we extract cross-attention maps in every cross-attention layer, smooth them with a Gaussian kernel, and then minimize the Frobenius norm to draw the subject outline and the scene contour. This localizes the appearance transfer within the subject-relative regions and introduces novel layouts from the vanilla Stable Diffusion, to improve generative diversity.**

## 2.3   Text-to-image personalization

Personalization aims to reason over specified subjects in assorted contexts. Textual Inversion [10] learns an embedding of a unique word to represent the specified subject. DreamBooth [23] fine-tunes the whole diffusion UNet to bind a unique identifier to the specified subject. Custom Diffusion [15] fine-tunes the key and value projection matrices in the cross-attention layers in the diffusion UNet. Encoder-based methods [11, 14, 30] fine-tunes the text encoder and potentially trains an image encoder or multi-modal transformer [16, 27] to encode example images of specified subjects into embedding that is leveraged in personalized generation. Our proposed Pick-and-Draw consistently improves identity reproduction of these personalization methods without harming the model diversity in a training-free one-shot manner.

We share inspiration from DisenBooth [5] and DETEX [1], which decouple the foreground and background regions in personalized generation. They aim to prevent the model from overfitting on a few samples during concept learning and fine-tune decoupled textual embeddings to separate subjects from irrelevant parts. In comparison, we impose constraints on intermediate features and cross-attention to correct the denoising path during inference time, which makes our approach training-free and model-agnostic.

## 3   Method

Our purpose is to perform training-free surgery in personalized diffusion models to boost identity consistency and generative diversity. Given a reference image $I_r$ that depicts a specified subject $s$

(*e.g.*, dog), our aim is to generate a personalized picture $I_p$ with a text prompt $P_p$ (*e.g.* "A dog in water"), which is consistent with the instance $I_r$ and conveys contextual semantics specified by $P_p$.

Current personalized models fail to reconcile both identity consistency and generative diversity, since they are prone to overfit on the given few subject images, inevitably reducing the generative latent space of diffusion models to a lower dimension. To alleviate the above issues, our core idea is to inject appearance information of the reference image and contextual priors from the original diffusion into the personalized generative process. The overall architecture is shown in Fig. 2. Our pipeline simulates a human painting process, we (1) adopt the intermediate feature set of the reference image as a palette, where we pick "colors" (*i.e.*, representative feature vectors that convey appearance information) to blend the new subject on the canvas, and (2) draw the outlines based on a generative template. We develop an appearance picking guidance and a layout drawing guidance for the above two procedures, respectively, and iteratively update the noisy latent via optimizing a designed energy function. The appearance picking guidance helps preserve the subject identity, and the layout drawing guidance ensures the generative diversity. We first discuss cross-attention saliency map extraction and selection strategy in Section 3.1, then introduce our two types of semantic guidance in Section 3.2 and Section 3.3 respectively.

### 3.1   Cross Attention Map Selection

Both the appearance and layout guidance rely on a saliency mask that highlights the subject-relative region. Previous works [12, 28,

31] show that cross-attention maps contain rich semantic and layout information. Similarly, we extract the subject-relevant cross-attention maps at each layer $l$ of the diffusion UNet:

$$\mathcal{A}_l = \text{softmax}(\frac{Q_l K_l^T}{\sqrt{d}}), \qquad (2)$$

where $Q_l$ is the query features projected from the image features, $K_l$ is the key features projected from the textual embedding with corresponding projection matrices, and $d$ is a scaling factor. We perform min-max normalization on these maps to acquire a set of normalized cross-attention saliency maps $\hat{\mathcal{A}}_{0:L} = \{\hat{\mathcal{A}}_0, \hat{\mathcal{A}}_1, ..., \hat{\mathcal{A}}_L\}$. We omit the timestep $t$ for simplicity.

Cross-attention maps of each layer contain different semantic information and highlight different regions of the image. Taking Stable Diffusion [22] as an example, the UNet consists of multiple up-blocks and down-blocks, with each block containing several cross-attention layers. We visualize the cross-attention maps corresponding to different layers in Fig. 3. Maps from deeper blocks have smaller resolutions. Specifically, we observe that (1) maps of 64×64 resolution are fine-grained and tend to outline the edges of all salient objects. They capture the high-frequency attributes, yet contain much background noise; (2) maps of 32×32 resolution are better aligned with the subject and highlight different regions, entailing richer semantic information; (3) maps of 16×16 resolution are coarse and well aligned, reflecting the approximate layout information of objects; (4) maps from encoder layers are mostly blended with more background noise, while maps from decoder layers better align with the subject layout, which conveys richer semantic and structural information.

Due to the disparate layout granularity and semantic information of cross-attention maps from each layer, we leverage different sets of maps for our proposed two types of semantic guidance. For appearance-picking guidance, we need to select the most representative activations to provide appearance cues. We choose the layers whose corresponding attention maps align well with the subject (or part of it). For layout drawing guidance, we need to introduce a novel image layout from an external prior. Note that the image layout not only includes the subject shape but also the context and background, and thereby we utilize attention maps from all layers.

## 3.2 Appearance Picking Guidance

Intuitively, we regard the generative feature set associated with the object region of the reference image as a palette, which provides essential visual cues for object appearance. To generate subjects that are consistent with the reference image, we assign the closest element in the palette to each generative feature vector within the subject-related region. We then minimize a transport distance to facilitate the diffusion model to gradually capture the appearance essence from the reference image during the generation process.

First, to obtain intermediate features of a reference image that depict the image content at each denoising step, we invert the reference image to the initial random noisy latent and feed it into the diffusion model to reproduce the denoising trajectory. We adopt Null-Text Inversion [17] as the inversion approach, which aligns the latent diffusion trajectory with the denoising trajectory by optimizing a null-text unconditional embedding in each step.

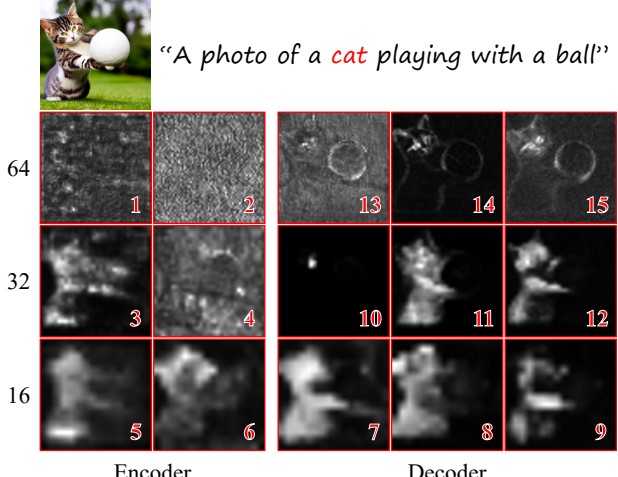

"A photo of a *cat* playing with a ball"

**Figure 3: Illustration of cross-attention maps extracted from different layers in the encoder and decoder of the UNet, numbered by inference order. Resolution is marked on the left.**

Second, we spot the local regions corresponding to the specific subject of reference image and generative image, and accordingly extract the saliency-aware feature sets for appearance transfer. Specifically, we apply hard masks $\mathcal{M}_l$ derived from normalized attention maps $\hat{\mathcal{A}}_l$ of each attention layer $l$ to highlight the subject-relative regions of the dense feature $\Psi_l$, and acquire saliency-aware visual features $\mathcal{V}_l$:

$$(\mathcal{M}_l)_{h,w} = \begin{cases} 1 & \text{if } (\hat{\mathcal{A}}_l)_{h,w} \geq \tau \\ 0 & \text{if } (\hat{\mathcal{A}}_l)_{h,w} < \tau \end{cases}, \qquad (3)$$
$$\mathcal{V}_l = \mathcal{M}_l \odot \Psi_l,$$

where $\tau$ is a threshold parameter and tuple $(h, w)$ represents a spatial entry of the attention map. We then extract the nonzero channels of $\mathcal{V}_l \in \mathbb{R}^{H_l \times W_l \times D_l}$ to obtain a feature set $\mathbf{V}_l = \{(V_l)_1, ..., (V_l)_n\}$, where $(V_l)_i$ is the $i$-th feature vector at the $l$-th layer and $n$ is the number of pixels within the salient region.

Subsequently, we search for an effective way to transfer the appearance essence from the reference palette to the generative canvas during the denoising process. Previous works provide inspiration that leverages Earth Movers Distance (EMD) to model the divergence between two feature distributions. Specifically, let $\mathbf{A} = \{A_1, ..., A_n\}$ and $\mathbf{B} = \{B_1, ..., B_m\}$ be two sets of $n$ and $m$ feature vectors, respectively. The EMD is formulated as:

$$\text{EMD}(\mathbf{A}, \mathbf{B}) = \min_{\mathbf{T} \geq 0} \sum_{ij} \mathbf{T}_{ij} \mathbf{C}_{ij},$$
$$s.t. \sum_j \mathbf{T}_{ij} = 1/m, \qquad (4)$$
$$\sum_i \mathbf{T}_{ij} = 1/n,$$

where $\mathbf{T}$ is the transport matrix which defines partial pairwise assignments, and $\mathbf{C}$ is the cost matrix which defines the distance between an element in $\mathbf{A}$ and an element in $\mathbf{B}$.

The EMD measures the cost of bidirectional optimal transport between two sets of features. However, directly minimizing EMD brings undesired artifacts and distorted subject posture. We believe it is due to the redundant bidirectional constraints of EMD. In the context of appearance transfer, we aim to align the generative pixels with those of the reference subject. Specifically, we minimize the unidirectional optimal transport cost from the generated image features $\mathbf{V}_l^{\text{gen}} = \{(V_l^{\text{gen}})_1, ..., (V_l^{\text{gen}})_m\}$ to the reference image features $\mathbf{V}_l^{\text{ref}} = \{(V_l^{\text{ref}})_1, ..., (V_l^{\text{ref}})_n\}$. We relax the EMD to a single constraint and define the Unidirectional Relaxed Earth Movers Distance (UREMD) as the appearance-aware loss:

$$\ell_{\text{app}} = \text{UREMD}(\mathbf{V}^{\text{ref}}, \mathbf{V}^{\text{gen}})$$
$$= \min_{\mathbf{T} \geq 0} \sum_{ij} \mathbf{T}_{ij} \mathbf{C}_{ij}, \tag{5}$$
$$s.t. \sum_i \mathbf{T}_{ij} = 1/n.$$

We aim to assign the closest element in $\mathbf{V}_l^{\text{ref}}$ to $V_j^{\text{gen}}$. In this manner, the aforementioned formulation is equivalent to:

$$\ell_{\text{app}} = \frac{1}{n} \sum_j \min_i \mathbf{C}_{ij}, \tag{6}$$

where we define the $(i, j)$-th entry $\mathbf{C}_{ij}$ of cost matrix $\mathbf{C}$ as the pairwise cosine distance between two feature vectors:

$$\mathbf{C}_{ij} = D_{\cos}(V_i^{\text{ref}}, V_j^{\text{gen}}) = 1 - \frac{V_i^{\text{ref}} \cdot V_j^{\text{gen}}}{\|V_i^{\text{ref}}\|\|V_j^{\text{gen}}\|}. \tag{7}$$

Simply put, at each step we find a one-to-one injection from the generated features to the reference features, and the mean UREMD between these two feature sets can be considered a metric evaluating overall subject appearance similarity. Optimizing the appearance-aware loss helps align the feature distributions of the reference image and generated image while avoiding excessive constraints on the generative layout, which is crucial for preserving the quality of the generated outcomes.

## 3.3 Layout Drawing Guidance

Previous works have successfully utilized cross-attention layout control on image editing [8, 12] and grounded generation [6, 31, 33]. The key idea is that cross-attention maps highlight the salient object-related region, specifying the shape, posture, and position of objects within the canvas. Since vanilla Stable Diffusion which is trained on massive image-text pairing datasets has been proven of impressive generative diversity, we aim to perform layout guidance to borrow its image prior and guide personalized generation. We regard the layout generated by vanilla Stable Diffusion as a template and draw the outline by imitating. In this way, the personalized model inherits strong generative priors of SD, ensuring the diversity of generative outcomes.

Unlike most works that aggregate the attention maps from each layer to a single saliency map, we leverage the cross-attention maps in all layers separately, as discussed in Section 3.1. Following Chefer et al. [4], we first smooth the attention maps with a Gaussian kernel to eliminate noisy perturbations, then calculate the distance of layer-wise attention maps between the generated image and the template image as layout-aware loss:

$$\ell_{\text{lay}} = \frac{1}{L} \sum_l \|G(\hat{\mathcal{A}}_l^{\text{gen}}) - G(\hat{\mathcal{A}}_l^{\text{temp}})\|_F, \tag{8}$$

where $G$ is the Gaussian kernel, $\|\cdot\|_F$ is the Frobenius norm and $L$ is the number of cross-attention layers.

By aligning the generative layout between the personalized model and the original diffusion model, we help the model inherit the strong generative priors, to synthesize diverse contexts according to different text conditions, and thus alleviate the model overfitting problem during generation. In practice, we also find that applying the layout-aware loss in the early steps of the denoising process helps pre-stabilizing the subject contour, providing a good initialization for performing the appearance guidance.

By combining the appearance picking guidance and layout drawing guidance together, we generate new subjects that highly align with the reference image among various contexts, pushing the trade-off between identity consistency and generative diversity to a new Pareto frontier. The overall loss function at step $t$ for personalized generation can be written as below:

$$\ell_t = \alpha_t \ell_{app} + \beta_t \ell_{lay}, \tag{9}$$

and the noisy latent $z_t$ is iteratively updated at step $t$ by

$$z_t \leftarrow z_t - \eta_g \nabla_{z_t} \ell_t, \tag{10}$$

where $\eta_g$ is the guidance ratio.

## 4 Experiments

### 4.1 Experimental Settings

**Dataset.** We conduct experiments on DreamBench [23], a dataset for text-to-image personalization performance evaluation. It consists of 30 subjects, including unique objects and pets such as backpacks, stuffed animals, dogs, cats, toys, etc. The subjects are separated into two categories, where 21 are objects and 9 are live subjects/pets. 25 text prompts are collected for testing generalization ability, including recontextualization and property modification. The evaluation suite requires generating 4 images for each subject and each prompt, amounting to a total of 3000 images.

**Evaluation Metrics.** We follow DreamBooth [23] and employ evaluation metrics of DINO [3], CLIP-I [20], and CLIP-T scores. DINO and CLIP-I scores are utilized to evaluate subject fidelity, while CLIP-T score is utilized to evaluate image-text fidelity. The DINO score is the average pairwise cosine similarity between the ViT-S/16 DINO embeddings of the generated and real images. The CLIP-I score is the average distance of pairwise CLIP ViT-B/32 image embeddings of the generated and real images. DINO score is a preferred metric due to its sensitivity in capturing variations among subjects within the same class. These two combined reflect identity consistency. Lastly, the CLIP-T score represents the average cosine similarity between the CLIP embeddings of text prompt and image. CLIP-I measures image-text alignment among various contexts, thereby reflecting the model's generative diversity.

**Implementation Details.** For our experiments, we set the guidance ratio to 10 and the binary mask threshold to 0.1. The weight of two losses alpha and beta are set to 10 and 100 respectively across all guidance steps. We perform guidance during the first 20 steps and

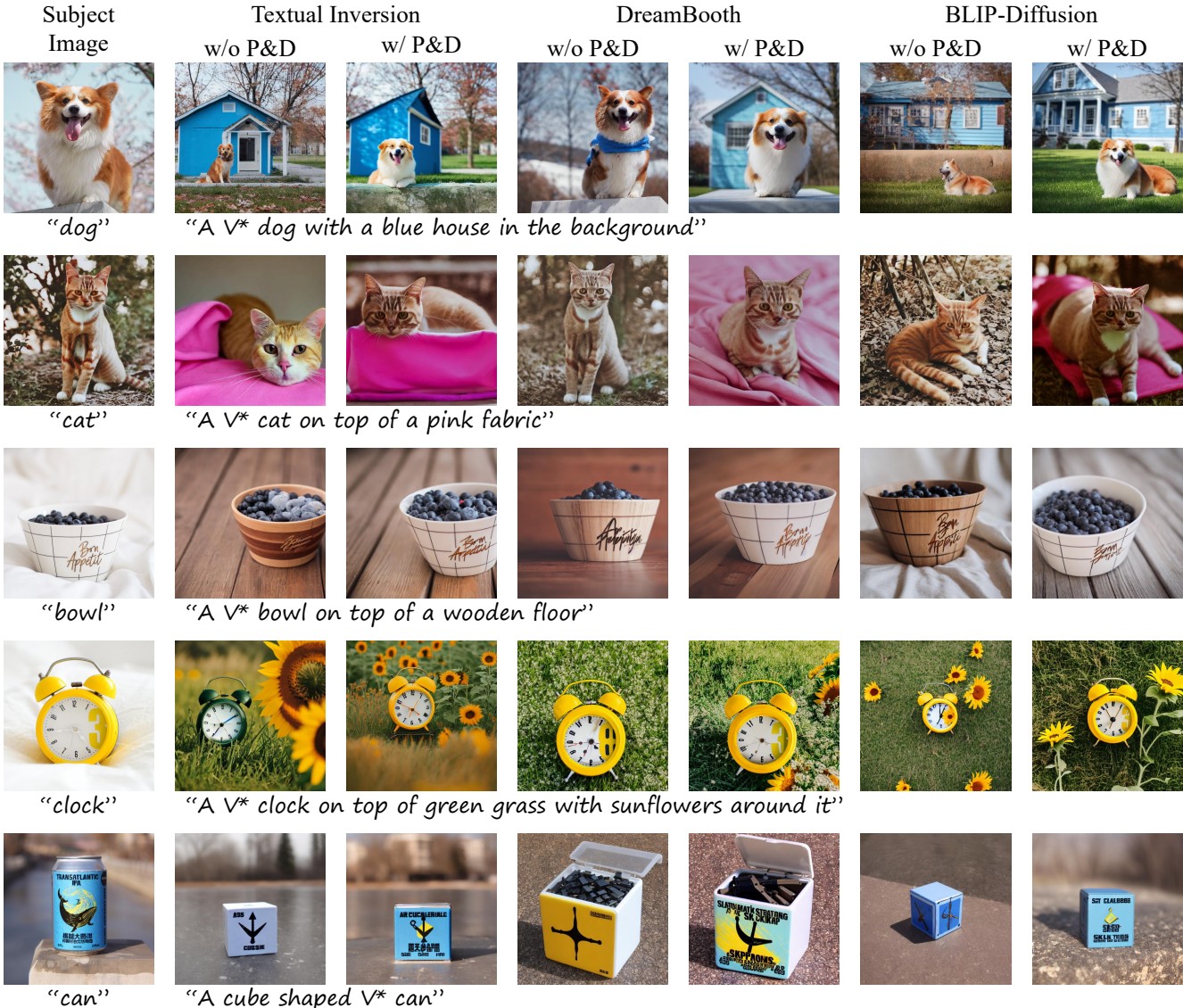

**Figure 4: Qualitative results on different baselines with and without Pick-and-Draw. The format of text prompt slightly differs across the three baselines and we choose the DreamBooth format for presentation.**

optimize 3 iterations per step. We use the same hyperparameters for all subjects and prompts without any extra tuning.

## 4.2 Main Qualitative Results

We provide qualitative comparisons of various text-to-image personalization methods including Textual Inversion [10], DreamBooth [23] and BLIP-Diffusion [16] with and without Pick-and-Draw in Fig. 4. See the Appendix for a detailed description of the baselines. We observe that Pick-and-Draw consistently improves both subject fidelity and image-text fidelity on all three baselines. Textual Inversion falls short in preserving identity; DreamBooth and BLIP-Diffusion better preserve the subject identity, but tend to

overfit and memorize the subject's pose and background, causing unsatisfactory alignment with the text prompt. Specifically, DreamBooth fails to generate the blue house (1st row) and the sunflowers (4th row) and overfits the cat image (2nd row), while BLIP-Diffusion memorizes the background of the forest (2nd row) and the white fabric (3rd row). All three methods fail to preserve appearance traits when performing geometric shape modification (5th row). Furthermore, many synthesized subjects exhibit inconsistencies in certain details.

In comparison, our proposed Pick-and-Draw (1) greatly enhances identity consistency, (2) improves generative diversity, and aligns better with the text prompt. The effectiveness of Pick-and-Draw

| Methods | P&D | DINO | CLIP-I | CLIP-T |
|---|---|---|---|---|
| Real Images (Oracle) | | 0.774 | 0.885 | - |
| Stable Diffusion | $\times$ | 0.320 | 0.504 | 0.339 |
| | $\checkmark$ | 0.552 (+0.232) | 0.641 (+0.137) | 0.335 (-0.004) |
| Textual Inversion | $\times$ | 0.568 | 0.664 | 0.252 |
| | $\checkmark$ | 0.627 (+0.059) | 0.745 (+0.081) | 0.263 (+0.011) |
| BLIP-Diffusion | $\times$ | 0.587 | 0.716 | 0.292 |
| | $\checkmark$ | 0.651 (+0.064) | 0.778 (+0.062) | 0.300 (+0.008) |
| DreamBooth | $\times$ | 0.616 | 0.739 | 0.297 |
| | $\checkmark$ | 0.696 (+0.080) | 0.790 (+0.051) | 0.303 (+0.006) |

**Table 1: Quantitative comparisons on DreamBench dataset. P&D are short for our proposed method Pick-and-Draw. Performance gains and losses are written in blue and red subscripts, respectively.**

can be attributed to the accurate semantics provided by appearance-picking guidance and layout-drawing guidance. The appearance picking guidance performs vigorous relaxed appearance transfer by calibrating the misalignment between the generated and reference appearance cues sets while the layout drawing guidance forces the personalized generation to an external layout, thereby mitigating the overfitting problem.

## 4.3 Comparisons on DreamBench Dataset

In Tab. 2, we reproduce three personalization methods and study the impact of Pick-and-Draw quantitatively on the DreamBench dataset. In particular, we include vanilla Stable Diffusion as an additional baseline and report DINO and CLIP-I scores of real images as the

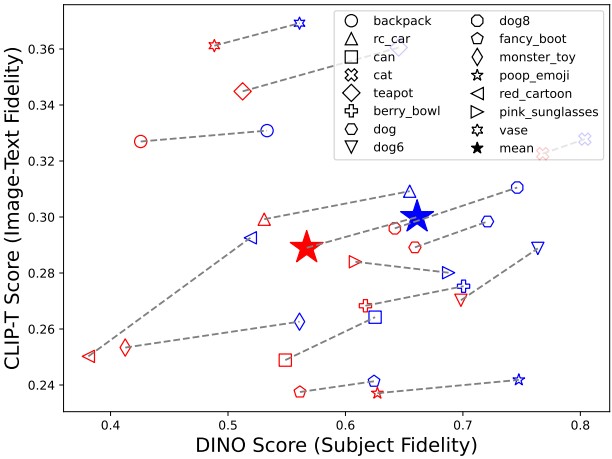

**Figure 5: Alignment metrics of BLIP-Diffusion before (red) and after (blue) applying Pick-and-Draw for sample subjects.**

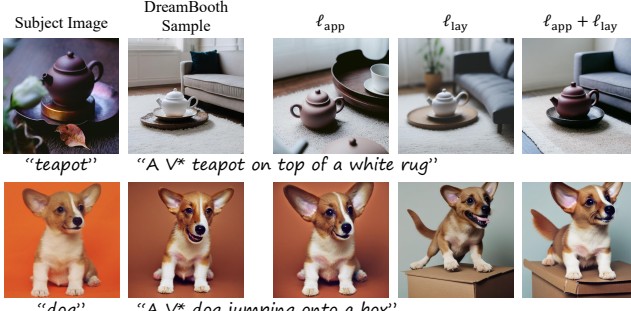

**Figure 6: Ablation for the effect of different losses, including the appearance-aware loss $\ell_{app}$, layout-aware loss $\ell_{lay}$ and both combined, conducted on DreamBooth.**

| | Textual Inversion | $16 \times 16$ | $32 \times 32$ | $64 \times 64$ |
|---|---|---|---|---|
| **DINO** | 0.568 | 0.582 | **0.627** | 0.593 |
| **CLIP-I** | 0.664 | 0.709 | **0.745** | 0.718 |
| **CLIP-T** | 0.252 | 0.241 | **0.263** | 0.259 |

**Table 2: Ablation results on different activation selection strategies for the appearance picking guidance, conducted on Textual Inversion. The best results are bold.**

performance upper bound. For every text prompt, we generate 4 images, summing up to a total of 3,000 images across all subjects.

The overall results are consistent with the qualitative findings, where Pick-and-Draw improves the performance of the three methods on all metrics. The remarkable performance gains on DINO and CLIP-I metrics can be attributed to the appearance picking guidance which performs accurate relaxed appearance transfer and greatly enhances identity consistency. The improved CLIP-T score can be attributed to the layout drawing guidance, which anchors the generated layout to an external prior and introduces more generative diversity, leading to better alignment with the text prompt. Additionally, in Fig. 5 we show per-subject metrics and observe that Pick-and-Draw significantly improves subject fidelity and improves image-text fidelity in most cases.

## 4.4 Ablation Study

**Impact of activation selection.** In Tab. 2, we quantitatively study the impacts of different activation selection strategies for the appearance picking guidance discussed in Section 3.2 for ablation. We choose Textual Inversion as the example baseline. We find that using activations of resolution $32 \times 32$ yields the best results on both subject fidelity and image-text fidelity. Specifically, activations of resolution $16 \times 16$ contain mainly layout information, thus unwanted layout leakage might happen and reduce the text prompt alignment. Activations of resolution $64 \times 64$ focus on fine-grained high-frequency details such as edges, which are not sufficient for local appearance transfer. In comparison, activations of resolution $32 \times 32$ encode rich semantic information and focus on different regions of the subject, facilitating the local appearance transfer and

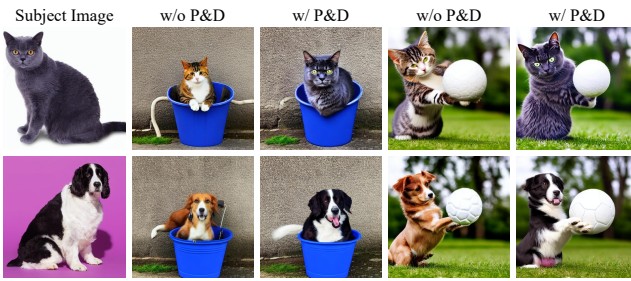

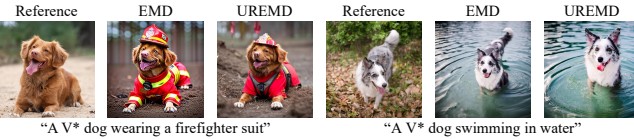

"A V* dog wearing a firefighter suit"    "A V* dog swimming in water"

**Figure 8: Ablation between EMD and UREMD on Dream-Booth. Please zoom in for a better view.**

subject and the reference subject share a similar shape and size. The bottom right of Fig. 7 is a failure case, where the size of the generated subject is inconsistent with that of the reference subject. We report the metrics of Stable Diffusion with and without Pick-and-Draw in Tab. 1. The subject fidelity is significantly improved at the cost of minor image-text fidelity, and this training-free approach demonstrates comparable overall performance to Textual Inversion. This may inspire further research on training-free single-image text-to-image personalization.

### 4.6 More Analysis

**Runtime cost.** Pick-and-Draw uses energy function-based optimization during inference and has similar time complexity to other such methods, including Self-Guidance [8], BoxDiff [33], and DragonDiffusion [18]. In Table 3, we compare the runtime between the methods mentioned above and our approach, tested on a single NVIDIA RTX-4090 GPU. Theoretically, the total number of forward and backward passes represents the runtime overhead. Our approach adopts a DDIM scheduler with 50 denoising steps, where we perform layout drawing guidance during the first 10 steps and appearance picking guidance between 10 and 20 steps. We optimize the noisy latent 3 times per step, which sums up to an extra 60 forward passes and 60 backward passes. Compared to the vanilla Stable Diffusion (50 forward passes), our method increases approximately two times overhead, which is worthwhile compared to repeated generation and cherry-picking (usually 10 times).

| SD | Self-Guidance | BoxDiff | DragonDiffusion | Pick-and-Draw |
|----|---------------|---------|-----------------|---------------|
| 7.9s | 35.1s | 31.6s | 25.7s | 28.9s |

**Table 3: Runtime cost comparison.**

## 5 Conclusion

In this paper, we propose Pick-and-Draw, a training-free semantic guidance approach for text-to-image personalization. We point out the prevalent overfitting issue of current methods: (1) they tend to memorize the few reference samples and struggle to generate diverse poses, views, and backgrounds of the subject; (2) they require careful hyperparameter tuning to achieve delicate results. To this end, we propose appearance-picking guidance and layout drawing guidance to boost performance for any personalized models with a single reference image. Qualitative and quantitative experiments demonstrate that Pick-and-Draw consistently improves identity consistency and generative diversity, pushing the trade-off between subject fidelity and image-text fidelity to a new Pareto frontier.

"in a bucket"      "playing with a ball"

**Figure 7: Visual results of Pick-and-Draw directly applying to vanilla Stable Diffusion.**

achieving the overall best result. This observation is consistent with the discussion in Section 3.1. Qualitative results are provided in the Appendix for intuitive visualization.

**Impact of two loss components.** We show visual results for comprehension of our proposed appearance-aware loss $\ell_{app}$ and layout-aware loss $\ell_{lay}$ in Fig. 6. We present two failure cases of DreamBooth, *i.e.* the attribute misbinding issue in the first row and the overfitting issue in the second row, and illustrate how the two losses work together to address these issues collectively. The $\ell_{app}$ facilitates local appearance transfer, which makes the generated subjects (the 3-rd column) more aligned with the reference image. However, the appearance transfer may be unbounded and incorrect (the 1-st row) and novel layouts are not introduced (the 2-nd row). The $\ell_{lay}$ (the 4-th column) constrains the appearance transfer within the subject region (the 1-st row) and introduces a novel layout (the 2-nd row), but without appearance picking guidance, the identity consistency of the new subject is not guaranteed. When combining the two losses for diffusion guidance, the generated outcomes (the last column) exhibit the best performance. Specifically, the appearance transfer is accurately bounded within the subject region, solving the attribute misbinding issue, while a novel layout is introduced with consistent subject identity, mitigating the overfitting problem.

**Choice between EMD and UREMD.** We make a qualitative comparison between EMD and UREMD in the appearance-picking guide in Figure 8. We notice that EMD loss may bring undesired artifacts and distorted subject posture, which confirms the discussion in Section 3.2. The intuition of our appearance-picking guidance involves treating the reference image as a palette and using its pixels to compose the synthesized subject. We aim to align the generative pixels with ones of the reference subject in a unidirectional manner, thus adopting UREMD instead of EMD.

We also discuss the impact of different guidance timestep selections. Quantitative results and analysis are provided in Appendix.

### 4.5 Results on Vanilla Stable Diffusion

We directly apply Pick-and-Draw on vanilla Stable Diffusion and observe surprisingly favorable results in some cases. The visual results are presented in Fig. 7. Without the strong subject prior of the fine-tuning-based personalization baselines, the appearance transfer still ensures identity consistency when the SD-generated

## Acknowledgments

This work was supported by National Natural Science Foundation of China: 62322211, 62336008, "Pionee" and "Leading Goose" R&D Program of Zhejiang Province (2024C01023), Key Laboratory of Intelligent Processing Technology for Digital Music (Zhejiang Conservatory of Music), Ministry of Culture and Tourism (2023DMKLB004).

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
