# OpenReview forum: "Pick-and-Draw: Training-free Semantic Guidance for Text-to-Image Personalization"
_acmmm.org/ACMMM/2024/Conference — MM2024 Poster_

### Official Review · Reviewer_rMsR · 2024-05-10

**Rating:** 3
**Confidence:** 3

**Summary:**

The author proposes Pick-and-Draw, a training-free semantic guidance approach to boost identity consistency and generative diversity for
personalization methods.

**Strengths:**

The method proposed by the author is effective.
The papaer is easy to follow.

**Limitations:**

1、There are some shortcomings in the experiment. The baseline model used is too old. The authors need to add more comparative experiments.
2、Lack of innovation. The method proposed by the author is only a simple application of prompt2prompt[1].
[1] Hertz A, Mokady R, Tenenbaum J, et al. Prompt-to-prompt image editing with cross attention control[J]. arXiv preprint arXiv:2208.01626, 2022.

**Suitability:**

2

---

### Official Review · Reviewer_cAVE · 2024-05-13

**Rating:** 1
**Confidence:** 4

**Summary:**

This paper proposed Pick-and-Draw, a training-free semantic guidance approach to boost identity consistency and generative diversity for personalization methods. This method consists of two components: appearance-picking guidance and layout-drawing guidance and can be applied to any personalized diffusion model and requires as few as a single reference image.

**Strengths:**

1. Good paper writing, solid work
2. The method is innovative and expandable to some extent, and can be combined with multiple personalized generation work to improve the original generation effect.
3. The details of the method are fully explored and analyzed in detail.

**Limitations:**

1. The paper only optimizes the performance on the basis of related work, which is not innovative enough.
2. The test case list is of a single type, with little comparative work and do not include the latest performance related work.
3. The complexity of the test text is not high enough
4. From the qualitative results, the ID fidelity of the generated image is still insufficient.

**Suitability:**

3

---

### Official Review · Reviewer_ofcN · 2024-05-23

**Rating:** 4
**Confidence:** 2

**Summary:**

The authors introduce Pick-and-Draw, an approach to text-to-image personalization that does not require training. They highlight a common problem with current methods: (1) they often memorize only a handful of reference samples and have difficulty generating a variety of poses, viewpoints, and backgrounds for the subject; (2) achieving nuanced results typically demands meticulous tuning of hyperparameters. To address these challenges, they present techniques for selecting appearances and drawing layouts to enhance the performance of personalized models using just one reference image.

**Strengths:**

1. Pick-and-Draw is a training-free plug-and-play approach for boosting text-to-image personalization, which is flexible and applicable.

2. Pick-and-Draw leverages subject-relevant cross-attention maps to incorporate the appearance information from the reference image and contextual priors from the original diffusion into the personalized generative process.

3. This paper is well-organized, with clear grammar and expression, making it easy to understand.

**Limitations:**

1. The novelty seems somewhat limited. Two loss function calculation methods are integrated into the diffusion process, which essentially constrains the attention maps to enhance generative capabilities.

2. How is generative diversity qualitatively manifested in Layout Drawing Guidance?

3. More state-of-the-art text-to-image personalization methods should be compared to demonstrate effectiveness, as only three are currently included.

**Suitability:**

3

---

### Official Review · Reviewer_XP3o · 2024-05-23

**Rating:** 5
**Confidence:** 4

**Summary:**

The paper proposes a training-free method to improve identity consistency and generative diversity in text-to-image models. The approach consists of two main components: appearance-picking guidance and layout-drawing guidance, which help generate diverse and contextually accurate images while preserving subject identity.

**Strengths:**

The paper is well-organized and written. The idea is novel and makes sense. The experiments are comprehensive and provide sufficient evidence to support the conclusions. The proposed method, Pick-and-Draw, effectively enhances identity consistency and generative diversity without requiring extensive retraining.

**Limitations:**

1. To find the local region corresponding to a specific subject in the image, the authors set a threshold to filter the pixels and generate a binary image. How was the threshold of 0.1 selected? If the threshold is too large or too small, what impact will it have on appearance picking and the final results?
2.  Lack of sensitivity analysis for different thresholds in the Ablation study.
3.  Despite claiming to be "training-free," the method involves multiple optimization steps that require additional computational resources. The paper should provide specific training times and configurations.

**Suitability:**

3

---

### Meta-Review · Area_Chair_Veyo · 2024-07-05

**Recommendation:** Accept (Poster)
**Confidence:** 4

**Metareview:**

This paper proposes Pick-and-Draw, a training-free semantic guidance approach for personalization methods, aiming to improve identity consistency and generative diversity. The approach consists of two components: appearance-picking guidance and layout-drawing guidance. It initially received 1 Weak Accept, 1 Borderline Accept, 1 Borderline Reject, and 1 Reject. Reviewers acknowledged the clarity of writing, flexible method design and the effectiveness of experiments. They also raised questions regarding comparative baselines, novelty, ablation studies, computational complexity, etc. The authors generally addressed those questions in the rebuttal. One reviewer raised score from Reject to Weak Reject, and the other three reviewers kept their scores. The AC finds the idea of this paper would be interesting to the community, and recommend to accept the paper.